# Bactericidal type IV secretion system homeostasis in *Xanthomonas citri*

**William Cenens**[1]*, **Maxuel O. Andrade**[2], **Edgar Llontop**[1], **Cristina E. Alvarez-Martinez**[3], **Germán G. Sgro**[1], **Chuck S. Farah**[1]*

**1** Departamento de Bioquímica, Instituto de Química, Universidade de São Paulo (USP), Av. Prof. Lineu Prestes 748, São Paulo, SP, Brazil, **2** Laboratório Nacional de Biociências, Centro Nacional de Pesquisa em Energia e Materiais, R. Giuseppe Máximo Scolfaro, Campinas, SP, Brazil, **3** Departamento de Genética, Evolução, Microbiologia e Imunologia, Instituto de Biologia, Universidade Estadual de Campinas (UNICAMP), Rua Monteiro Lobato, Campinas, SP, Brazil

* w.cenens@gmail.com (WC); chsfarah@iq.usp.br (CSF)

**Data Availability Statement:** All relevant data are within the manuscript and its Supporting Information files.

## Abstract

Several *Xanthomonas* species have a type IV secretion system (T4SS) that injects a cocktail of antibacterial proteins into neighbouring Gram-negative bacteria, often leading to rapid lysis upon cell contact. This capability represents an obvious fitness benefit since it can eliminate competition while the liberated contents of the lysed bacteria could provide an increase in the local availability of nutrients. However, the production of this Mega Dalton-sized molecular machine, with over a hundred subunits, also imposes a significant metabolic cost. Here we show that the chromosomal *virB* operon, which encodes the structural genes of this T4SS in *X. citri*, is regulated by the conserved global regulator CsrA. Relieving CsrA repression from the *virB* operon produced a greater number of T4SSs in the cell envelope and an increased efficiency in contact-dependent lysis of target cells. However, this was also accompanied by a physiological cost leading to reduced fitness when in co-culture with wild-type *X. citri*. We show that T4SS production is constitutive despite being downregulated by CsrA. Cells subjected to a wide range of rich and poor growth conditions maintain a constant density of T4SSs in the cell envelope and concomitant interbacterial competitiveness. These results show that CsrA provides a constant though partial repression on the *virB* operon, independent of the tested growth conditions, in this way controlling T4SS-related costs while at the same time maintaining *X. citri*'s aggressive posture when confronted by competitors.

## Author summary

*Xanthomonas citri* is a member of a family of phytopathogenic bacteria that can cause substantial losses in crops. At different stages of the infection cycle, these cells will encounter other bacterial species with whom they will have to compete for space and nutrients. One mechanism which improves a cell´s chance to survive these encounters is a type IV secretion system that transfers a cocktail of antimicrobial effector proteins into other Gram-negative bacteria in a contact-dependent manner. Here, we show that this system is

**Funding:** This work was supported by FAPESP - Fundação de Amparo a Pesquisa do Estado de São Paulo (http://www.fapesp.br/) through post-doctoral scholarships to WC (2015/18237-2), GGS (2014/04294-1), and EL (2019/12234-2) and research grants to CSF (2011/07777-5 and 2017/17303-7). The funders had no role in study design, data collection and analysis, decision to publish, or preparation of the manuscript.

**Competing interests:** The authors have declared that no competing interests exist.

constitutively produced at a basal level, even during low nutrient conditions, despite representing a significant metabolic burden to the cell. The conserved global regulator, CsrA, provides a constant, nutrient-independent, repression on the production of T4SS components, thereby holding production costs to a minimum while at the same time ensuring *X. citri*'s competitiveness during encounters with bacterial rivals.

## Introduction

Type IV secretion systems (T4SS) are large multiprotein nanomachines spanning both the inner and outer membrane of many Gram-negative bacterial species [1]. The best described functions of T4SSs are the delivery of the T-DNA of *Agrobacterium tumefaciens* [2,3], their roles in bacterial conjugation [4] and their involvement in delivering virulence factors to mammalian cells by several pathogenic species [5]. T4SSs found in Gram-negative bacteria are made up of over 100 subunits of 12 different proteins (VirB1 to VirB11 plus VirD4) with a total size of over 3 MDa each [1] though some bacterial species carry systems with extra components and are much larger in size [6,7]. Maintaining and expressing large operons and producing the amino acids required to assemble the proteins they encode present a significant investment in terms of energy and raw materials for a cell [8–10]. Given the high cost secretion systems would have on cell physiology, it is not surprising that their production is often restricted to specific conditions, where they will be most needed. For example, expression of the *A. tumefaciens* T4SS is dependent on pH, monosaccharides, phosphate and specific phenolic compounds released by wounded plant tissue [11–14]. Similarly, plasmid-borne *tra* genes, encoding a T4SS and other components of the conjugation machinery, are only produced during specific conditions and often only in a small portion of the population [15]. Other examples include the *Brucella suis* T4SS produced inside acidic phagocytic vacuoles of macrophages [16] and the *Ehrlichia ruminantium* T4SS whose genes are induced during iron starvation [17]. This strict environmentally-dependent regulation is also common in other secretion systems; for example, the *Vibrio cholera* type VI secretion system (T6SS) is induced during high cell densities on chitinous surfaces [18], the *Xanthomonas citri* T6SS is specifically induced in the presence of amoeba [19], the *Shigella flexneri* type III secretion system (T3SS) is tightly regulated by oxygen levels [20] and T3SS expression in *Xanthomonas* species is induced upon interaction with their plant hosts [21,22].

*X. citri* is a phytopathogen that causes citrus canker, a disease which can lead to significant losses in citrus fruit production [23]. Previously, our group has characterized the interbacterial killing activity of a T4SS in *X. citri* and showed that this strain actively transfers a cocktail of antibacterial effector proteins into neighbouring Gram-negative cells in a contact-dependent manner [24,25]. More recently, we also described the antibacterial killing of a similar T4SS with its unique antibacterial effectors in the opportunistic pathogen *Stenotrophomonas maltophilia* [26]. Despite the importance of T4SSs in interbacterial competition, little is known concerning the regulation of these Xanthomonadales-like T4SSs (X-T4SSs).

Microarray data of an *X. citri* strain harbouring a mutation in the global regulator CsrA (also called RsmA) indicated its involvement in the regulation of over a hundred genes, including the *virB* operon that encodes the T4SS proteins VirB1-11 [27]. CsrA is a pleiotropic regulator linked to the genetic changes during stationary phase growth, biofilm formation, gluconeogenesis and virulence [28]. CsrA acts by binding specific mRNA loops in 5′ untranslated regions containing the canonical 5′-GGA-3′ motif [29,30]. In some cases, these interactions stabilize the mRNA leading to increased expression, as for example has been observed

for the *hrpG* mRNA in *X. citri* [27]. More often, these CsrA-binding loops encompass the ribosome binding site, in which case binding of CsrA inhibits translation [31]. Although several other means of CsrA regulation exist [28], the majority of interactions lead to a repression of protein production [32]. CsrA/RsmA is regulated by two important small RNAs, CsrB and CsrC (in *Escherichia coli*) or RsmY and RsmZ (in *Pseudomonas aeruginosa*), which contain several high affinity CsrA binding loops that effectively titrate CsrA [33,34]. Production of these small RNAs in *E. coli* is controlled by several regulatory pathways, including the BarA/UvrY two-component system that responds to molecules such as formate and acetate, the catabolite repression pathway mediated by cAMP-CRP and the stringent response governed by RelA and SpoT-mediated production of (p)ppGpp [28]. Furthermore, direct regulation of CsrA copy numbers in *E. coli* is achieved by five different promoters using at least two different sigma factors [35].

X. citri CsrA is very similar to CsrA from *E. coli* and *P. aeruginosa* (>77% identical), albeit the *X. citri* protein has a 9 residue C-terminal extension. Detailed knowledge of CsrA, its targets and its regulation in *Xanthomonas* species, whose genomes do not code for CsrB and CsrC homologs, is limited [36]. Nonetheless, some studies have shown phenotypic alterations in a CsrA deletion strain reminiscent of known CsrA phenotypes in *E. coli* and *P. aeruginosa*, such as reduced virulence, increased biofilm formation and increased glycogen production [37,38] and direct RNA binding studies have also confirmed the affinity of *X. citri* CsrA for the canonical 5′-GGA-3′ motifs [27].

In this work, we show that CsrA from *X. citri* represses the *virB* operon and that the removal of this repression has a measurable fitness cost. However, CsrA repression is incomplete, and so production of the *virB* products continues at a controlled basal level, maintaining a constant density of T4SSs in the cell envelope during different growth conditions. This sustained and energetically affordable aggressive posture contributes to *X. citri* competitiveness and survival.

## Results

### CsrA regulates the *virB* operon by binding to the 5′UTR of *virB7*

Based on transcription start site analysis data for *Xanthomonas campestris* [39] and further observations made in the Sequence Read Archive for *X. campestris* and *X. citri* (https://www.ncbi.nlm.nih.gov/sra), the Xanthomonas *vir* locus contains two main transcription start sites (TSSs) (Fig 1A). The presence of these TSSs in *X. citri* was confirmed by 5′RACE analysis (S1 Fig). The first TSS is located 303 nucleotides upstream of the *virD4* start codon and a second TSS is located 249 nucleotides upstream of the *virB7* start codon (Fig 1A and S1 Fig). Thus, both *virD4* and *virB7* contain large upstream regions, with the upstream region of *virD4* containing an open reading frame that encodes a conserved protein of unknown function. Although an open reading frame can also be detected between *virD4* and *virB7* (nucleotide sequence shown in Fig 1A), it lacks a canonical ribosome binding site and its translated product is of very low sequence complexity and is not conserved in the protein databases. The organization of the open reading frames in the *virB* locus (*virB7-virB11* followed by *virB1-virB6*) plus three more downstream open reading frames (*xac2611*, *xac2610* and *xac2609*) suggest that they may be expressed as a polycistronic mRNA (Fig 1A and S2A Fig). To test this hypothesis, *X. citri* RNA was purified and reverse-transcribed to produce cDNA which was then used as a template in overlapping PCR assays. Employing a set of nine specific primer pairs, we were able to show that a single polycistronic mRNA (approximately 13.7 kbp in length) is produced that encompasses all the genes from *virB7* to *xac2609* but does not extend beyond *xac2609* (S2B Fig). We note that *xac2609* and *xac2610* code for a Xanthomonadaceae type IV

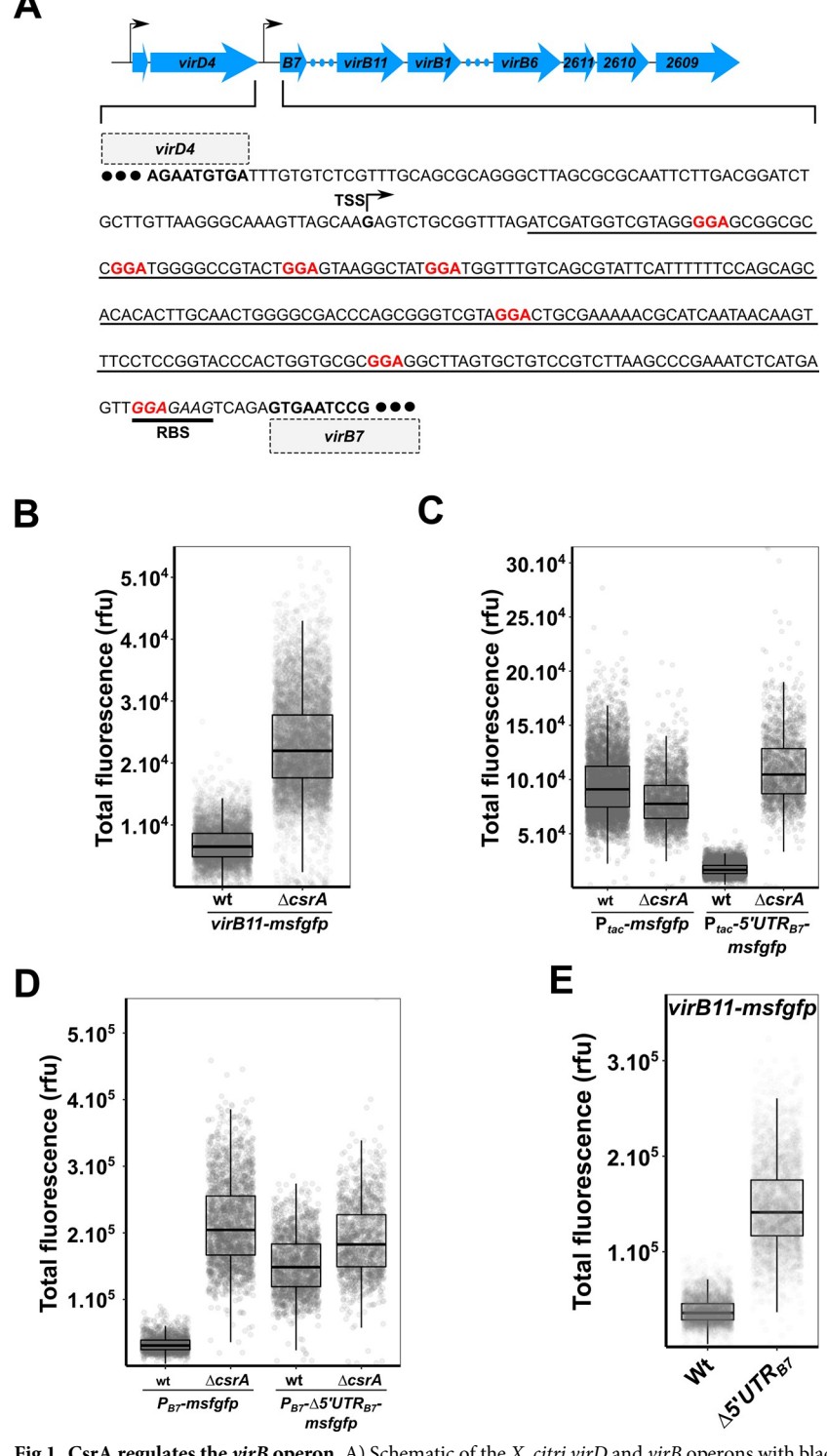

**Fig 1. CsrA regulates the *virB* operon.** A) Schematic of the *X. citri virD* and *virB* operons with black arrows indicating positions of the transcription start sites, blue arrows indicating the open reading frames and blue dots representing genes not shown. The intergenic sequence between *virD4* and *virB7* genes is shown and the *virB7* transcription start site (TSS) is indicated. The 5´-GGA-3´ motifs (at positions +31, +42, +57, +70, +141, +194 and +238/RBS, respectively) are indicated in red and the predicted ribosome binding site (RBS) is underlined with a thick black line. The sequence that is deleted in the *Δ5´UTR_{B7}* strains is underlined with a thin black line. B) Single-cell msfGFP fluorescence levels of the *X. citri virB11-msfgfp* transcriptional reporter in *X. citri* wild-type and *ΔcsrA* strains.

Production of msfGFP increased on average 3.3-fold in the *ΔcsrA* strain (N$_{wt}$ = 4887 cells and N$_{ΔcsrA}$ = 6263 cells, from a single representative culture each). C) Single-cell msfGFP fluorescence levels of the *amy::P$_{tac}$-msfgfp* and *amy::P$_{tac}$-5 ´UTR$_{B7}$-msfgfp* reporters in *X. citri* wild-type and *ΔcsrA* strains. For the *amy::P$_{tac}$-5´UTR$_{B7}$-msfgfp* strain: N$_{wt}$ = 6862 cells and N$_{ΔcsrA}$ = 1671 cells, from 2 separate cultures each. For the control *amy::P$_{tac}$-msfgfp* strain: N$_{wt}$ = 5534 cells and N$_{ΔcsrA}$ = 2495 cells, from 2 separate cultures each. D) Single-cell msfGFP fluorescence levels of the *msfgfp* reporter that substituted the entire structural *virB* operon (with the *msfgfp* start codon placed at the position of the *virB7* start codon) in *X. citri* wild-type and *ΔcsrA* strains with an intact *virB* promoter (*P$_{B7}$-msfgfp*) or a *virB* promoter in which the *5´UTR$_{B7}$* was deleted (*P$_{B7}$-Δ5´UTR$_{B7}$-msfgfp*). For *X. citri* cells carrying the *ΔvirB::P$_{B7}$-msfgfp* reporter: N$_{wt}$ = 1495 cells and N$_{ΔcsrA}$ = 2156 cells, from 2 separate cultures each. For *X. citri* cells containing the *ΔvirB::P$_{B7}$-Δ5´UTR$_{B7}$-msfgfp* reporter: N$_{wt}$ = 1748 cells and N$_{ΔcsrA}$ = 1304 cells, from 2 separate cultures each. E) Single-cell msfGFP fluorescence levels from the *X. citri virB11-msfgfp* reporter strain with and without the genomic deletion of the *5´UTR$_{B7}$* (underlined region in part A). N$_{wt}$ = 3397 cells and N$_{Δ5´UTR}$ = 3099 cells, from 4 separate cultures. Tukey box-and-whisker plots in parts B-E: black central line (median), box (first and third quartiles) and whiskers (data within 1.5 interquartile range).

secretion system effector (X-Tfe$^{XAC2609}$) previously shown to be secreted by the X-T4SS and its cognate immunity protein (X-Tfi$^{XAC2610}$) [24].

The list of CsrA-regulated genes in *X. citri* from a microarray dataset includes several of the *virB* genes [27]. We therefore further scrutinised the *virB* 5´UTR (upstream of *virB7* start codon, from here on referred to as *5´UTR$_{B7}$*). This analysis revealed several 5´-GGA-3´ motifs (Fig 1A) that could be CsrA binding sites when present in a stable loop structure [30]. To test this hypothesis, a transcriptional *msfgfp* (encoding for the monomeric super folder Green Fluorescent Protein) fusion was constructed downstream of the genomic copy of *virB11* (*X. citri virB11-msfgfp*), located at the centre of the *virB* operon. Single-cell fluorescence analysis showed that upon deleting *csrA* in *X. citri virB11-msfgfp*, msfGFP production from the *virB* operon is upregulated 3.3-fold (Fig 1B). In order to confirm the direct regulation of CsrA on the *5´UTR$_{B7}$*, this region was cloned in between the P$_{tac}$ promoter (constitutive in the *lac* negative *X. citri* strain) and *msfgfp*, after which a single copy of this construct was integrated into the α-amylase gene (*amy*; *xac0798*) in both the *X. citri* wild-type and *ΔcsrA* strains. Measurements of the fluorescence levels in thousands of individual cells show that CsrA is capable of repressing msfGFP production (5.8-fold decrease) from the P$_{tac}$ promoter only when the *5´UTR$_{B7}$* is present, but not in its absence (Fig 1C). Similar results (3-fold decrease) were obtained by comparing β-glucuronidase activity from *amy::P$_{tac}$-5´UTR$_{B7}$-gus* reporters in *X. citri* wild-type and *ΔcsrA* strains (S3B Fig). These results indicate that CsrA exerts its repression in a mechanism that is, at least in part, independent of the *virB* promoter. Expanding these observations, we replaced the entire structural *virB* operon (Fig 1A) with *msfgfp*, to produce the *X. citri ΔvirB::P$_{B7}$-msfgfp* strain (with the *msfgfp* start codon in the exact position of the *virB7* start codon). Deletion of *csrA* in this strain produced a similar increase in msfGFP production (Fig 1D). Importantly, the removal of the *5´UTR$_{B7}$* in this strain (*X. citri ΔvirB:: P$_{B7}$-Δ5´UTR$_{B7}$-msfgfp*) caused msfGFP levels to increase in the wild-type strain (Fig 1D). This increase in msfGFP production in the absence of *5´UTR$_{B7}$* was only slightly lower than that observed in the *X. citri ΔvirB::P$_{B7}$-msfgfp ΔcsrA* and *X. citri ΔvirB::P$_{B7}$-Δ5´UTR$_{B7}$-msfgfp ΔcsrA* strains (Fig 1D), further confirming CsrA regulation mediated by the *5´UTR$_{B7}$*. We note that the *ΔcsrA* background has an elaborate effect on *X. citri* physiology (for instance, cultures display cellular flocculation and decreased cell-sizes) that could account for the subtle differences in msfGFP production observed between strains lacking *csrA* with or without the *5´UTR$_{B7}$* in Fig 1D. Next, the construction of a genomic deletion of the *5´UTR$_{B7}$*, while keeping all other *virB* genes intact, resulted in a 3.9-fold increase in expression levels in the *X. citri Δ5´UTR$_{B7}$ virB11-msfgfp* reporter strain (Fig 1E). An electrophoretic mobility shift assay (EMSA) confirmed the direct *in vitro* binding of CsrA to the *5´-UTR$_{B7}$* (S4 Fig). Taken together, the results presented so far suggest that CsrA regulates *virB* operon expression by binding to the *5*

´UTR$_{B7}$, most probably by preventing translation and/or destabilizing the *virB* transcripts. Fig 1A highlights the GGA motifs in the *5′UTR$_{B7}$* at positions +31, +42, +57, +70, +141, +194 and within the putative ribosome binding site at position +238. To determine if one or more of these sites could contribute to this regulation we introduced mutations in the *5′UTR$_{B7}$* region of the *amy::P$_{tac}$-5′UTR$_{B7}$-gus* reporter in the *X. citri* wild-type strain. S3C Fig shows that deleting the first 72 nucleotides of the *5′UTR$_{B7}$* that contains the first 4 GGA motifs, does not significantly affect GUS activity. The remaining 3 GGA motifs were mutated to GAA (strains 141-GAA, 194-GAA and RBS-GAA). A ~1.7-fold increase in GUS activity was observed for the 141-GAA strain while no significant change in gene reporter activity was observed for the 194-GAA strain (S3C Fig). A ~1.7-fold increase was also observed for a strain containing both 141-GAA and 194-GAA mutations. As could be expected, the RBS-GAA mutation, either on its own or in combination with 141-GAA, resulted in a drastic decrease in β-glucuronidase activity (S3C Fig), most likely due to compromised ribosome binding and translation initiation. Thus, we have evidence that part of CsrA's regulatory effect could be due to its binding to the GGA motif at position 141 of the *virB* transcript. Since the effect of this mutation (1.7-fold) was not as significant as that observed for the *ΔcsrA* strain (3-fold), we cannot rule out the possibility that CsrA is binding to other sites in the *5′UTR$_{B7}$*, for example the GGA motif within the RBS.

Since the results shown in Fig 1 are based on the fluorescence intensity of cells expressing an *msfgfp* reporter gene, we sought to observe the direct effect of CsrA and the *5′UTR$_{B7}$* on *vir* mRNA and protein levels. Fig 2A shows quantitative real-time PCR experiments in which the relative levels of transcripts for all the open reading frames from *virD4* to *xac2609* in the *ΔcsrA* and *Δ5′UTR$_{B7}$ virB11-msfgfp* strains are presented with respect to the levels observed in the wild-type strain. Deletion of *csrA* resulted in 8 to 16-fold increases in the amount of message observed for *virD4*, all of the *virB* open reading frames as well as the three downstream reading frames (*xac2611*, *xac2610* and *xac2609*), consistent with microarray data previously reported for the *ΔcsrA* strain [27]. Deletion of the *5′UTR$_{B7}$* also resulted in significant increases in the amount of message containing the first ten open reading frames of the *virB* operon (*virB7-virB11* plus *virB1-virB5*) but not for the last four (*virB6*, *xac2611*, *xac2610*, *xac2609*) (Fig 2A). As expected, *virD4* transcript levels did not increase in the *5′UTR$_{B7}$ virB11-msfgfp* background (Fig 2A). We performed western blots to analyse the levels (in total cell lysates) of the VirB7, VirB8, VirB9 and VirB10 components of the X-T4SS, coded by the first four open reading frames of the *virB* operon, and of X-Tfe$^{XAC2609}$, coded by the last open reading frame. Fig 2B and 2C and S5 Fig show that the *virB11-msfgfp* reporter strain and the *X. citri* wild-type strain (1.0 fold-change baseline) have very similar levels of all five proteins, while the deletion of *csrA* in either wild-type or *virB11-msfgfp* backgrounds results in 2.2 to 7.3-fold increases in all of these proteins. Deletion of the *5′UTR$_{B7}$* in the *virB11-msfgfp* reporter strain also resulted in 2.2 to 5.4-fold increases in the detected levels of VirB7, VirB8, VirB9 and VirB10 while X-Tfe$^{XAC2609}$ levels increased by a more modest factor of 1.8-fold (Fig 2B and 2C). Thus, a variety of experimental approaches (Figs 1 and 2) point to a direct role of CsrA and the *5′UTR$_{B7}$* in the repression of the *virB* operon.

## Removal of CsrA/5′UTR$_{B7}$ repression on the *virB* operon increases T4SS numbers and bacterial killing

We then asked whether the absence of the *5′UTR$_{B7}$* would result in an increased number of assembled X-T4SSs in the bacterial envelope and a consequent enhancement in interbacterial killing efficiency. For these assays we took advantage of an *X. citri virB10-msfgfp$_{TL}$* translational fusion strain [40] in which the periplasmic VirB10 component has been replaced by a

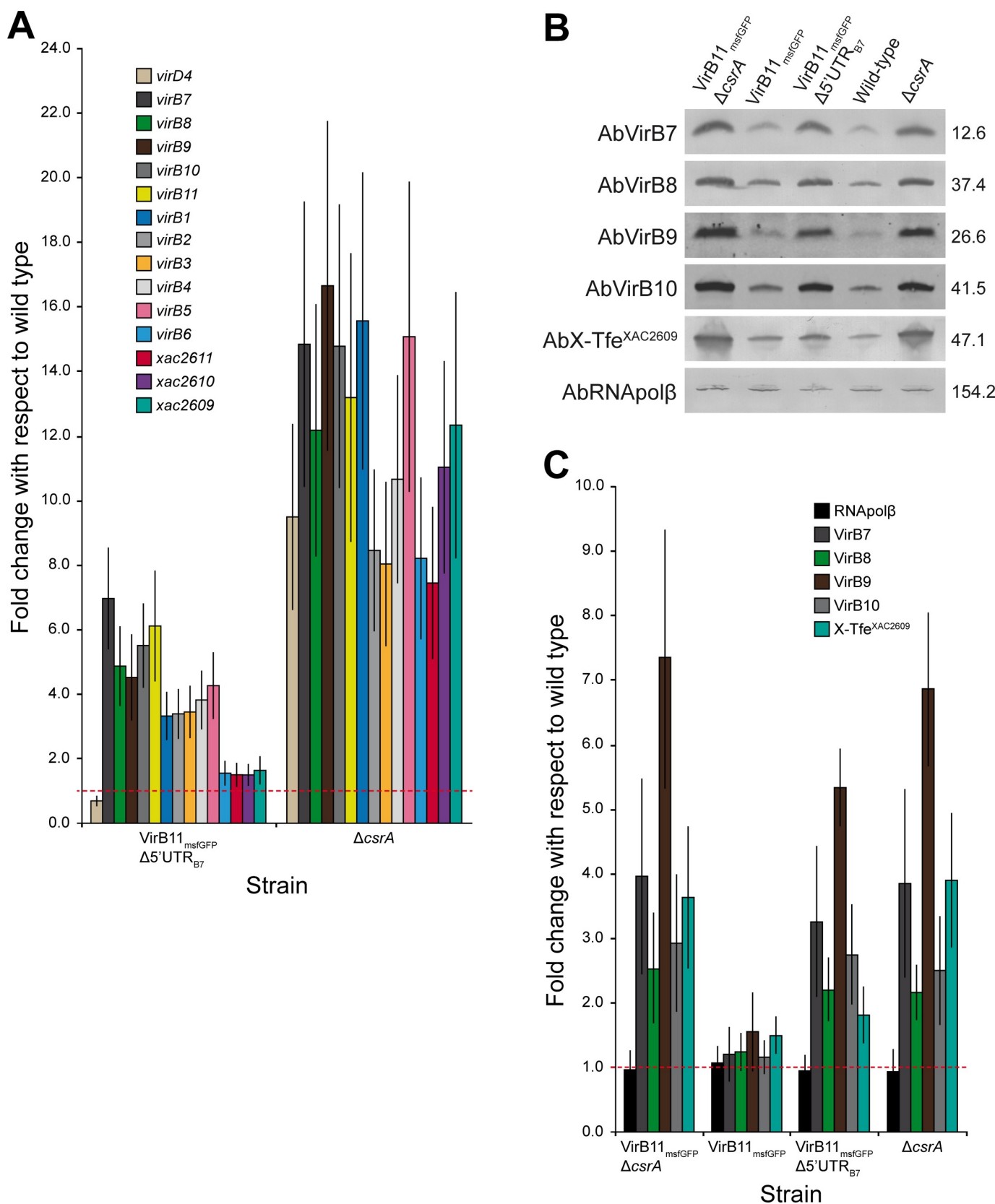

**Fig 2. Quantitative analysis of X-T4SS gene expression and protein levels in X. citri.** A) X-T4SS gene expression analysis in different genomic backgrounds. Quantitative real-time polymerase chain reaction (RT-qPCR) of representative genes within the *X. citri* X-T4SS *virD* and *virB* operons. Vertical bars represent the mean value from four RT-qPCR runs (two biological and two technical replicates for each gene). Vertical lines represent the standard deviations. The 16S gene was used as an internal reference housekeeper gene. The *X. citri* wild-type strain (fold change = 1.0 represented by the dashed horizontal red line) was used as the reference sample. B) Western blot band intensity as a measurement of protein accumulation in *X. citri* total cell lysates. Assays were performed using polyclonal antibodies (Ab) against specific *X. citri* X-T4SS structural components or effector (X-Tfe[XAC2609]) in different *X. citri* strains (see Materials and Methods). To adjust for small differences in sample loading and transfer, monoclonal antibodies against the β subunit of RNA polymerase (AbRNApolβ) were used. Experiments were repeated five times with similar results. Full western blot membranes are presented in S5 Fig. Theoretical molecular weight (in kDa) for each mature protein is shown at the right. C) Quantitative analysis of protein accumulation. Band intensities from membranes, as in B, were normalized to the values of *X. citri* wild-type strain (fold change = 1.0 represented by the dashed horizontal red line) for each assayed protein (see Materials and Methods). Bars represent the mean value from at least four experiments (minimum two biological replicates) and vertical error bars represent the standard deviation.

VirB10-msfGFP chimera. Since each T4SS contains 14 copies of VirB10, assembled T4SSs can be observed as fluorescent periplasmic foci and counted (see Materials and Methods and [40]). Deleting the *5´UTR$_{B7}$* in the *X. citri virB10-msfgfp$_{TL}$* genetic background resulted in a 2.6-fold increase in the number of fluorescent T4SS foci that were counted per cell (Fig 3A). We note that the higher density of T4SSs in this strain leads to a more crowded periplasm, making it more difficult to clearly separate individual foci, leading to an underestimation of total number of X-T4SSs. Therefore, the calculated 2.6-fold increase should be considered a lower limit.

As could be expected, a greater number of T4SSs in the *X. citri Δ5´UTR$_{B7}$virB10-msfgfp$_{TL}$* strain also increased the efficiency with which *X. citri* lyses *E. coli* cells in a quantitative LacZ mediated CPRG-cleavage assay (Fig 3B). Using the slopes of the curves in the CPRG-cleavage assays as a measure of killing efficiency (see Materials and Methods and [40]), this strain kills 2.1-fold more efficiently than the *X. citri virB10-msfgfp$_{TL}$* strain (Fig 3B). However, these results also show that under these conditions, removal of CsrA repression was not necessary to observe X-T4SS dependent *E. coli* lysis in the CPRG assays (Fig 3B), in agreement with previously published spot assays and CFU-based competition assays, all performed with wild-type *X. citri* strains [24]. To test whether we could identify conditions in which X-T4SS-mediated killing would be inhibited or enhanced, we tested *E. coli* lysis efficiencies at 18˚C or 28˚C, at pH 6.0, 7.0 and 8.0, in the absence of $Fe^{3+}$ and using different carbohydrate sources (glucose, sucrose or starch; Fig 3C). All the tested conditions led to clearly detectable T4SS-dependent lysis of *E. coli* cells with only small to moderate variations (+/- 35%) in killing efficiencies (Fig 3C). Taken together, these observations indicate that the observed levels of msfGFP signal from the *X. citri virB11-msfgfp* reporter strain (Fig 1A, 1B and 1E) and the discrete numbers of T4SSs in the cell envelope (Fig 3A) present in the wild-type background, represent the baseline expression and production levels of T4SS components and that these levels, all repressed from what they would otherwise be in the absence of CsrA, are sufficient to maintain X-T4SS production and efficient killing of neighbouring target cells.

## X-T4SS dependent interbacterial killing is maintained during growth under scarce nutrient conditions

Considering the expected high energetic cost of producing T4SSs and the previously described responsiveness of *csrA* regulons to nutrient input [28], we decided to test whether lowering the nutrient contents of the growth media would change T4SS-dependent killing efficiencies. To accomplish this, *X. citri* cultures were transferred to media containing normal or reduced levels of sucrose and/or casamino acids (the only nutrient sources present in the defined media) and grown overnight followed by an additional nine hours in fresh media for each culture to fully adapt to the conditions. These cultures were then subjected to a quantitative CPRG-cleavage assay under standardized conditions (see Materials and Methods for details). Fig 3D shows

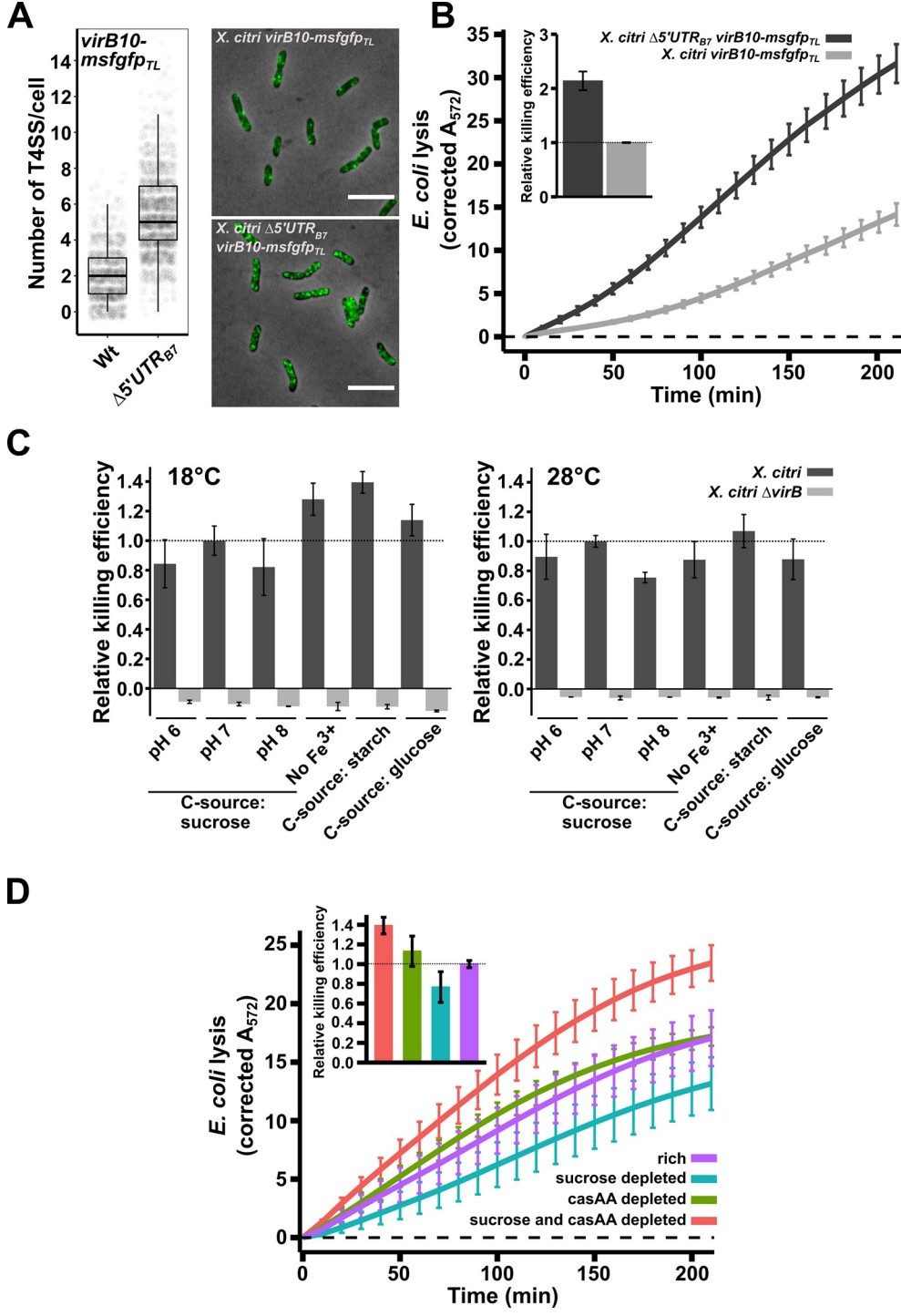

**Fig 3. X-T4SS production and inter-species bacterial killing is constitutive under incomplete repression by CsrA.**
A) Quantification of single fluorescent foci of X-T4SSs containing the VirB10-msfGFP chimera. The genomic deletion of the *5′UTR_B7* in the *X. citri virB10-msfgfp_TL* strain increased the number of X-T4SSs per cell 2.6-fold ($N_{wt}$ = 1361 cells and $N_{Δ5′UTR}$ = 2062 cells, from 3 separate cultures). Due to occasional overlap of the greater number of fluorescent foci in the *Δ5′UTR_B7* strain, the number of T4SS per cell may be underestimated in this strain. B) Quantitative CPRG cleavage-based killing assay using the same *X. citri virB10-msfgfp_TL* and *X. citri Δ5′UTR_B7 virB10-msfgfp*_TL cultures analysed in part A. Corrected $A_{572}$ values are given in arbitrary units (see Materials and Methods). The inset shows the normalized slope value of the linear part of the depicted curves relative to the wild-type strains (N = 4 separate cultures with 2 technical repeats each). C) Quantitative CPRG cleavage-based killing assays for

wild type *X. citri* cells grown in different conditions (dark grey bars). Killing efficiencies were evaluated in defined media containing different carbohydrate sources (0.2% sucrose, 50 μg/ml starch or 0.2% glucose), pH 6.0, 7.0 and 8.0, lack of $Fe^{3+}$ and at different temperatures (18˚C and 28˚C). Values represent the slope of the linear part of the $A_{572}$ curves as described in part B and normalized relative to the standard condition at pH 7.0. As a control, killing efficiency of a T4SS-deficient mutant (*X. citri ΔvirB::P_{B7}-msfgfp*; light grey bar) was assessed after growth under the same conditions (N = 4 separate cultures with two technical repeats). D) Quantitative CPRG cleavage-based killing assays from *X. citri* cells grown at different nutrient levels. Killing efficiencies of wild type *X. citri* strains are maintained during growth in defined media containing either 0.2% or 0.01% sucrose and/or casamino acids (casAA). Inset shows the slope value of the linear part of the depicted curves normalized with respect to the killing curve from the *X. citri* cultures grown in rich media (relative killing efficiency = 1.0, horizontal line). N = 5 separate cultures with 4 technical repeats. Error bars in panels B, C and d indicate the standard deviation. Horizontal dashed lines in panels B and D represent the zero-line obtained for the background signal unchallenged (non-lysed) *E. coli* cultures grown in parallel during each experiment.

that *X. citri* grown in nutrient-scarce culture media continues to sustain T4SS-dependent lysis of *E. coli* cells. *X. citri* cultures with limited access to sucrose have slightly reduced killing efficiencies (23% decrease) while cells with limited access to both casamino acids and sucrose presented slightly elevated (39% increase) killing efficiencies (Fig 3D). S1 Movie presents a time-lapse video of a mixed culture of *X. citri* and *E. coli* cells growing on thin agarose pads containing casamino acid-depleted media (AB medium + 0.2% sucrose + 0.01% casamino acids) for over 30 hours in which many events of *E. coli* cell lysis are observed upon contact with *X. citri* cells. For comparison, S2 Movie presents *X. citri* killing *E. coli* cells under standard nutrient concentrations (AB medium + 0.2% sucrose + 0.2% casamino acids), and also shows clear cell lysis upon contact with *X. citri* cells. No *E. coli* lysis is observed in these experiments when X-T4SS-deficient *X. citri* or *S. maltophilia* strains are employed [24,26,40].

## Constant but incomplete CsrA-mediated repression of the *virB* operon during different growth conditions

Figs 1E and 3A, show that removing the *5′UTR_{B7}* leads to both increased msfGFP production and T4SS assembly. In order to test how CsrA regulation impacts protein production from the *virB* operon, we decided to look more closely at VirB production in the presence and absence of the *5′UTR_{B7}* under different growth conditions. To do this, we began by comparing cytoplasmic msfGFP production from *X. citri virB11-msfgfp* and *X. citri Δ5′UTR_{B7} virB11-msfgfp* strains. Both strains were grown simultaneously in media with sucrose, glucose, glycerol or starch as carbohydrate sources or in media depleted for casamino acids and/or sucrose. Cultures were grown overnight in the different media, diluted and grown for additional 6 hours in fresh media before msfGFP content was registered by fluorescence microscopy.

   Mean msfGFP production from the *X. citri virB11-msfgfp* strain remains fairly similar during different inputs of carbohydrates and even in the absence of a carbohydrate source (Fig 4A, white boxplots). Reducing casamino acid concentrations, however, increases VirB production by 60% on average (both in combination with 0.2% or 0.01% sucrose). This increase in expression levels seems to be independent of CsrA regulation since similar increases in msfGFP production were detected when using the *X. citri Δ5′UTR_{B7} virB11-msfgfp* strain (Fig 4A, dark grey boxplots). In fact, when comparing the relative increases observed upon the deletion of the *5′UTR_{B7}*, it appears that repression mediated by the *5′UTR_{B7}* leads to a reduction of VirB protein production by, on average 3.8-fold (± 0.4) over all conditions tested (Fig 4A, fold-changes are indicated above each pair of boxplots). As such, it seems that CsrA represses the *virB* operon to an extent that is largely independent of the carbohydrate source or casamino acid availability.

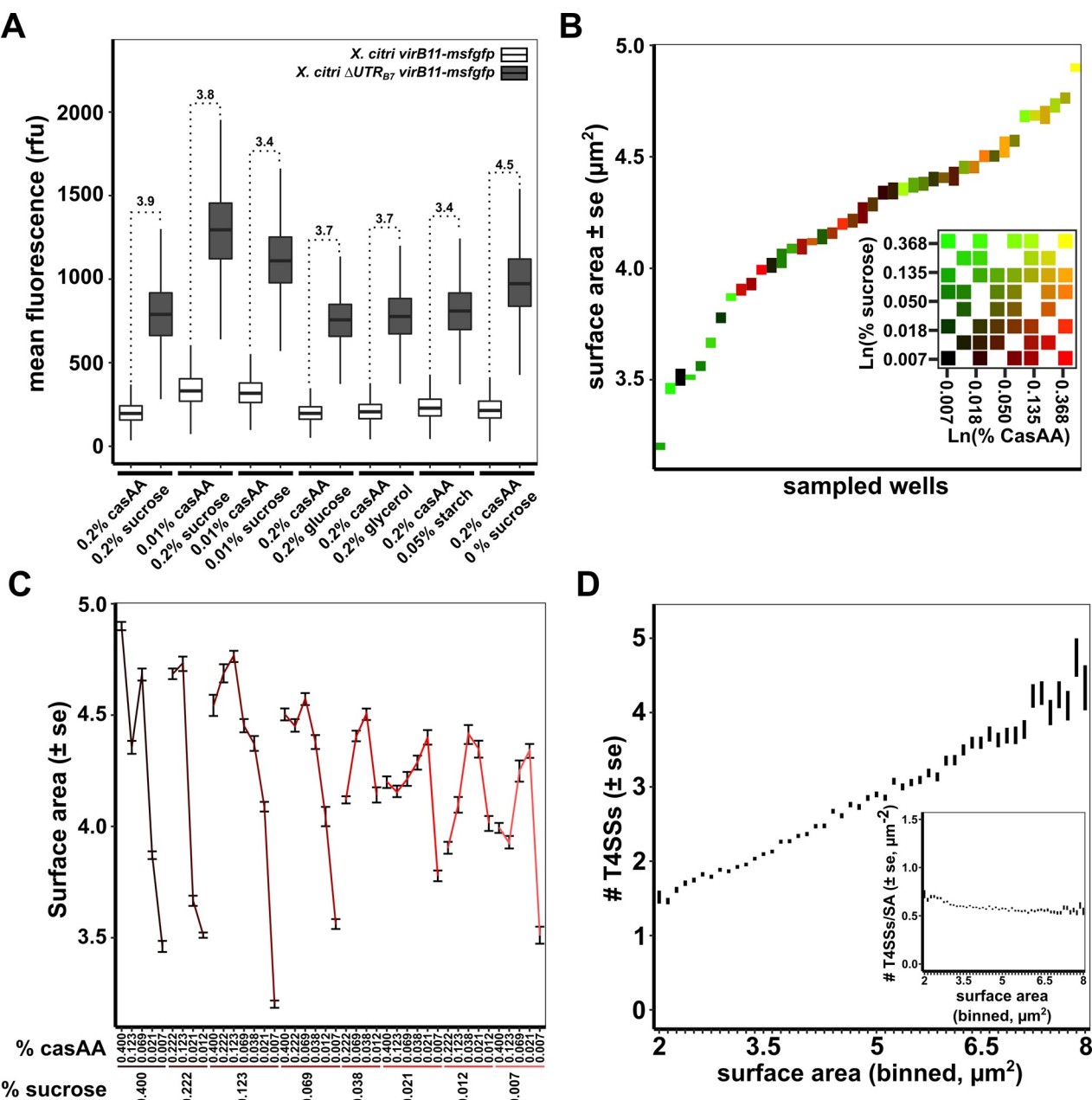

**Fig 4. X-T4SS production and density in the cell envelope under different nutrient conditions.** A) msfGFP fluorescence levels from *X. citri virB11-msfgfp* and *X. citri Δ5´UTR_B7 virB11-msfgfp* strains grown in AB defined media containing either sucrose, glucose, glycerol, starch or no carbohydrate source (0% sucrose) in combination with 0.2% casamino acids (casAA) or in AB defined media containing 0.01% casamino acids with 0.2% or 0.01% sucrose. Mean cytoplasmic msfGFP production from the *X. citri virB11-msfgfp* transcriptional fusion strain in the presence (white boxplots) or absence (grey boxplots) of the *5´UTR_B7* are represented as Tukey box-and-whisker plots for each media composition with black central line (median), box (first and third quartiles) and whiskers (data within 1.5 interquartile range). The *Δ5´UTR_B7 virB11-msfgfp vs virB11-msfgfp* ratio of the mean fluorescence levels is given above the Tukey plots for each growth condition. On average 6018 cells were sampled per condition for each strain, with a minimum of 2162 and a maximum of 16351 cells from two independent cultures. B) Surface area of *X. citri* cells sampled from separate cultures grown with different combinations of sucrose and casamino acid concentrations. Cultures are ordered according to the average cell surface area observed for each growth condition. The inset shows the 8x8 matrix range of sucrose and casamino acid concentrations used. In both the graph and the inset, each growth condition is colored differently. C) Surface area of *X. citri* cells as a function of sucrose and casamino acid concentrations. The data is the same as in part B but organised with respect to sucrose and casamino acid concentrations. D) Average number of T4SSs per cell versus cell surface area from all the cells grown in the different conditions presented in panels B and C. Pearson correlation r = 0.34, as calculated for all data points. *Inset*: Density of T4SSs in the cell envelope (number of T4SS per surface area, T4SS/SA) versus the surface area. Vertical bars in parts B, C and D represent the standard error of the mean. In total, 69412 cells were registered for the data in parts B, C and D.

## Homeostasis of X-T4SS density in the cell envelope over a wide range of nutrient availability

Spurred by the increased expression of the *virB* operon during decreased inputs of casamino acids, we set out to assess the number of assembled T4SS over a wide range of casamino acid and sucrose concentrations. For this, an 8 by 8 matrix of wells containing defined media with varying nutrient levels was created in a 96-well plate. With sucrose (rows) and casamino acids (columns) ranging from 0.4% to 0.007% in a 1.8x dilution series. After 24 hours of growth and an additional 5 hours of growth after a 4-fold dilution in the same but fresh media, *X. citri virB10-msfgfp$_{TL}$* cells were sampled and immediately imaged by fluorescence microscopy, registering both cell dimensions and the number of T4SS foci present per cell. In total, 42 different conditions (Fig 4B inset) were sampled over two separate experiments. Fig 4B and 4C show that the different nutritional inputs in each well result in a range of cell sizes, represented by their average surface areas and reveal a general correlation of reduced cell size with reduced casamino acid or sucrose availability. Importantly, plotting the number of T4SSs versus cellular surface area shows a linear increase in T4SS numbers with increasing surface area (Fig 4D, Pearson correlation *r* = 0.34, as calculated for all data points). Furthermore, this positive correlation in turn reflects an almost constant average T4SS density (T4SS/surface area) in the cell envelope with a subtle increase observed for the smallest cells (Fig 4D, inset). This is in line with the observation of subtly increased msfGFP production in the *X. citri virB11-msfGFP* transcriptional reporter at low casamino acid concentrations (Fig 4A). These results suggest that during the cell-cycle, when the surface area gradually increases until cell division, T4SSs are added continuously. Therefore, it seems that an *X. citri* population maintains the density of its T4SSs within a specific range under a variety of nutritional conditions. The number of T4SSs relative to surface area (T4SS density) could be an important factor in determining the probability that an *X. citri* cell is able to successfully transfer effectors into a neighbouring target cell during interbacterial competition.

## X-T4SS overproduction in a Δ5´UTR$_{B7}$ background has an impact on X. citri physiology and leads to reduced growth speeds

Since *Δ5´UTR$_{B7}$* cells present a roughly 4-fold greater expression from the *virB* operon (Fig 4A) and kill with approximately twice the efficiency as wild-type cells (Fig 3B), we asked whether this potentially advantageous feature could be counter-balanced by the inherent metabolic cost of X-T4SS production. We therefore set up a co-culture experiment to test whether overproduction of X-T4SSs in the *Δ5´UTR$_{B7}$* background leads to a detectable growth disadvantage in *X. citri*. For this, we took advantage of the difference in msfGFP production levels between *X. citri virB11-msfgfp* and *X. citri Δ5´UTR$_{B7}$ virB11-msfgfp* (Figs 1E and 4A) to sort wild-type and overproducing *Δ5´UTR$_{B7}$* cells by fluorescence microscopy (S6 Fig). This mitigated the need to introduce different antibiotic resistance markers in the genome that could on their own lead to subtle physiological differences. Additionally, the strains used here are genetically very closely related since they were obtained from the same recombination events leading to either the wild-type or mutant *Δ5´UTR$_{B7}$* allele (see Materials and Methods). Separate cultures of different single colonies of each of the strains were grown in defined AB media before being mixed and diluted in a 1:1 ratio into AB media supplemented with 0.2% sucrose and 0.01% casamino acids. Cultures were diluted regularly to avoid saturation and fluorescence microscopy images of thousands of cells were obtained at different time points over a period of approximately one week. Fig 5A shows the results of two distinct experiments, using a total of seven separate cultures, where we observed a 19% and 13% average decrease in the *X. citri Δ5´UTR$_{B7}$ virB11-msfgfp* cell population relative to the wild-type *X. citri virB11-msfgfp* cell

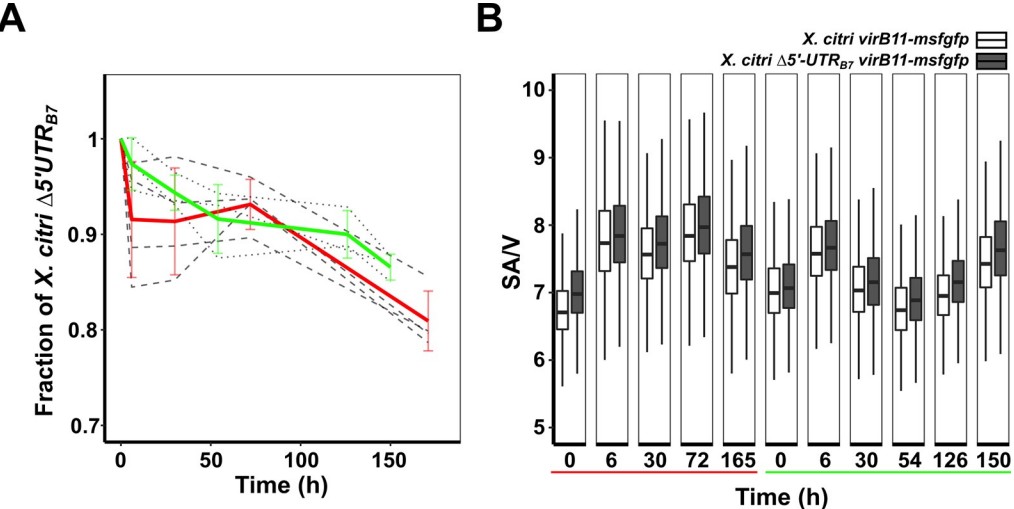

**Fig 5. X-T4SS overproduction leads to a detectable physiological cost for *X. citri*.** A) Co-culture experiment between *X. citri virB11-msfgfp* and X. citri *Δ5´UTR_{B7} virB11-msfgfp cells* in liquid medium. Cells were imaged by fluorescence microscopy and separated by the difference in msfGFP production levels as seen in Fig 1E. Results from two distinct experiments with three (dotted lines and green mean values) or four (dashed lines and red mean values) separate cultures are shown. The fraction of *X. citri Δ5´UTR_{B7} virB11-msfgfp* cells relative to the wild-type *X. citri virB11-msfgfp* cells at the end-points were 0.87 ± 0.01 (green) and 0.81 ± 0.03 (red). B) Comparison of surface area to volume ratios of *X. citri virB11-msfgfp* and *Δ5´UTR_{B7} virB11-msfgfp* for cells collected at different times for all seven cultures shown in part A. At each time point, an average of 3871 cells were analysed for each culture (minimum: 1061 cells, maximum: 9582 cells).

population. Additionally, analysis of the cell sizes of the two strains at several time points over a one-week period revealed that cells with the *Δ5´UTR_{B7}* background consistently have a slightly larger surface area over volume ratio (SA/V; Fig 5B), indicative of smaller cell sizes. The SA/V ratio has been suggested to be a measure of the physiological state of rod-shaped bacterial cells and is thought to be set by the availability of nutrients [41,42]. The observed average size reduction (increase in SA/V) for *X. citri Δ5´UTR_{B7} virB11-msfgfp* cells could be attributed to the increased metabolic cost of nutrients consumed in T4SS production and/or to stress induced by the greater number of T4SSs in the cell envelope. This experiment reveals that T4SS production has a measurable cost and illustrates the importance of balancing expenses with gains from increased aggressiveness during interbacterial competition.

## Discussion

We show here that the regulation of the *virB* operon in *X. citri*, coding for the structural proteins of its bacteria-killing X-T4SS, is under control of the global regulator CsrA which binds to sites in the 5´ untranslated region of the polycistronic mRNA initiating at the *virB7* gene (Fig 1). The removal of the *5´UTR_{B7}* leads to an increase in production of *virB* encoded proteins, which in turn leads to an increase in number of T4SSs present in the cell envelope and greater interbacterial killing efficiency (Fig 3A and 3B). Despite CsrA repression over the *virB* operon, we have not yet found any growth condition that would greatly decrease nor increase T4SS production and T4SS-dependent killing by *X. citri* cells (Figs 3C, 3D and 4A). Therefore, the removal of CsrA repression does not seem to be required to induce production of the T4SS. Rather, CsrA repression maintains a discrete number of T4SSs in the cell envelope over a range of different growth conditions tested (Figs 3 and 4).

Several factors can be imagined to be influencing the efficiency with which an *X. citri* cell can mount a successful contact-dependent attack. Firstly, an X-T4SS needs to be present at the contact interface between the attacking *X. citri* cell and the target rival cell. After a successful contact, X-T4SS effectors need to be translocated through the X-T4SS. This is dependent on both the availability of effectors and, importantly, ATP to power the secretion. For example, the subtle decrease in killing efficiencies in the sucrose-depleted conditions (Fig 3D) might stem from reduced energy levels, since sucrose would be the main carbohydrate fed into the glycolysis and citric acid metabolic pathways. Furthermore, depleting nutrients also leads to much smaller cell sizes compared to the cells grown in rich media [41]. This reduction in cell size in itself could influence killing-efficiencies by increasing the probability of small cells contacting larger *E. coli* cells when in co-culture on a solid surface, since smaller *X. citri* cells will have a more close-packed arrangement next to *E. coli* cells [43], increasing the probabilities of successful T4SS contact. Expanding this line of reasoning, a unit mass of cells formed during growth in reduced nutrient conditions would have a higher contact probability as the same mass of cells formed during growth in rich conditions, because of the former's greater number of single cells and greater SA/V ratio for each individual cell. Thus, the subtle increase in T4SS density with lower nutrient inputs (Fig 4B) and the concomitant smaller cell sizes, could both be responsible for the slight increments in killing efficiency during growth in media depleted in both casamino acids and sucrose (Fig 3D).

The complex regulatory features governing CsrA production and activity, including negative autoregulation, have been proposed to act to keep CsrA levels and activity relatively stable with greatly reduced cell-to-cell variability, making CsrA an ideal regulator for homeostatic responses [28,35]. Of note, an ancestral CsrA homolog has been shown to act as a homeostatic control agent during flagella morphogenesis in the Gram-positive *Bacillus subtilis* [44]. As such, CsrA could integrate diverse signals that relay information regarding the nutritional environment of the cell and subsequently stabilise its own activity to ensure a constant regulatory effect on its target transcripts. In light of this model for CsrA function, our experiments do indeed indicate that in *X. citri*, this protein acts to reduce production from the *virB* operon with roughly the same repressing power over a wide range of growth conditions (Fig 4A). Recently, it was reported that CsrA (RsmA) represses all three type VI secretion systems from *P. aeruginosa*, abolishing translation under non-inducing conditions [45]. Affinity of RNA loops for CsrA can differ by several orders of magnitude [46] and so some CsrA-mRNA associations will be very sensitive to fluctuations in mRNA levels and CsrA availability while others remain insensitive. As such, CsrA-mRNA affinities could be tuned so as to constitutively stabilise translation from one transcript at basal levels (such as for the *virB* operon) and at the same time ensure that translation from other transcripts is only derepressed by a specific trigger; for example, by a specific condition that changes the mRNA structure. Several genetic circuits leading to different outcomes are possible, but for sake of general discussion, we note that it is unlikely that CsrA availability varies greatly under different conditions, since this would be hard to reconcile with the simultaneous control of the hundreds of transcripts through which CsrA exerts its pleiotropic effects [27,32].

The removal of CsrA-based repression of T4SS production results in a two-fold increase in interspecies bacteria killing efficiency (Fig 3B) which could be beneficial under certain circumstances. However, the production of a T4SS has its costs: the maintenance and transcription of an approximately 13 kb locus, multiple rounds of translation to produce the over 100 subunits that need to be transported to the cell envelope and assembled into a single system of over 3 MDa in size [1]. Since the maintenance of even a single gene and the production of the amino acids to build up its protein product has a measurable cost [8–10], it was not surprising that we observed a reduction in the fitness of the *X. citri* strain in which CsrA-based repression was

removed (Fig 5). These differences in fitness levels were detected by scrutinizing many thousands of single cells in well-controlled laboratory settings. An *X. citri* strain with a hyper-active *virB* operon could perhaps have an advantage to survive hostile encounters with competitive species in *ex-planta* niches. However, we strongly suspect that in the field, and over many cycles of interactions with its host during which the bacterial population can expand rapidly in the absence of competing bacterial species, the advantage of CsrA-attenuation of the *virB* operon will manifest itself.

Interbacterial competition is increasingly being shown to be crucial for fitness, survival [47,48] and structuring of bacterial populations [49,50] and possibly contribute to bacterial evolution by uptake of DNA from lysed target cells [51]. The continuous expression and activity of the X-T4SS under different growth conditions, further illustrates the importance of interbacterial killing and the benefits that are accompanied with it. The control of *X. citri* X-T4SS production by CsrA seems to guarantee constant densities of this nanomachine at the bacterial surface, high enough to provide protection against rival bacteria, but not so high as to represent a metabolic burden. In this way, a balance is maintained that maximizes long-term *X. citri* survival in the varied natural environments it encounters during its life cycle, both within and outside of its plant host.

## Materials and methods

### Bacterial strains, media and culturing

All strains used are listed in S1 Table. For all experiments, strains were grown in defined AB media containing 15 mM $(NH_4)_2SO_4$, 17 mM $Na_2HPO_4$, 22 mM $KH_2PO_4$, 50 mM NaCl, 0.1 mM $CaCl_2$, 1 mM $MgCl_2$ and 3 μM $FeCl_3$, at pH 7.0, supplemented with 10 μg/mL thiamine and 25 μg/mL uracil and varying concentrations of carbohydrate sources and casamino acids as described in the text. For cloning purposes standard lysogeny broth (5g/l yeast extract, 5 g/l NaCl, 10 g/l tryptone and 15 g/l agar) and 2xYT (5 g/l yeast extract, 10 g/l NaCl, 16 g/l tryptone and 15 g/l agar) were used. For counterselection, sucrose plates were used (5 g/l yeast extract, 10 g/l tryptone, 60 g/l sucrose). Standard incubations were performed at 28˚C in 24-well plates using 1.5 ml culture media or in 96-well plates with 200 μL culture media with shaking at 200 rpm. In general, after a first overnight growth period in 2xYT medium, cells were transferred at a 100-fold dilution into AB defined media for a second overnight growth to synchronise growth. Cells were then diluted in fresh media and, after 4- to 6-hour growth, imaged by microscopy. In case of experiments involving different growth media compositions, cultures were inoculated once more at a 100-fold dilution in the appropriate AB media composition for overnight growth and a final re-inoculation in fresh media with dilutions ranging from 2-fold to 100-fold, depending on the overnight attained optical densities. Note that cultures grown under different nutrient conditions attained different densities after overnight growth. Care was therefore taken to dilute each culture in its appropriate media such as to obtain adequate numbers of cells for experimental assays but at the same time avoiding saturation of the faster growing cultures. After a final 5- to 9-hour growth period, cells were either imaged with fluorescence microscopy or subjected to competition assays.

### Cloning of constructs for genomic insertions and deletions

All primers, plasmids and strains used for cloning and PCR verifications together with brief description of the constructions are listed in S1 Table. Genomic deletions and insertions in the *X. citri* genome were all constructed using a two-step allelic exchange procedure [52]. For this, 500 to 1000 base pair-sized fragments up- and downstream from the region of interest were amplified using a high-fidelity polymerase (Phusion, Thermo Scientific) and cloned into the

pNPTS138 vector either by traditional restriction digest cloning (NEB and Thermo Scientific) or by Gibson assembly (NEB). The resulting plasmid was used to transform the appropriate *X. citri* strain by electroporation (2.0 kV, 200 Ω, 25 μF, 0.2 cm cuvettes; Bio-Rad)[53]. A first recombination event was selected for on LB plates containing 50 μg/ml kanamycin. Transformants were streaked for single colonies on kanamycin plates whereafter several single colonies of the merodiploids (Kan$^R$, Suc$^S$) were streaked on sucrose plates selecting for a second recombination event creating either a wild-type (reversion) or mutant allele. After confirmation of the loss of the kanamycin resistance cassette together with *sacB*, a PCR was performed using primers that hybridize outside of the homology regions to identify the target allele. Strains containing the wild-type alleles (revertants created at an equal rate during the second recombination event) were also stored and used as controls for their respective mutants. For the insertion of the $P_{tac}$-5′UTR$_{B7}$-*msfGFP* and $P_{tac\_}$*msfGFP* reporter constructs (or $P_{tac}$-5′UTR$_{B7}$-*gus* reporter constructs, see below) into the *amy* gene of *X. citri* we used the pPM7G plasmid [54]. *X. citri* cells were electroporated with the pPM7G-based constructs pPM7G-$P_{tac\_}$*msfGFP* and pPM7G-$P_{tac}$-5′UTR$_{B7}$-*msfGFP* (S1 Table) and selected for on LB plates containing 50 μg/mL kanamycin. Integrity of the *virB* operon in these strains was confirmed by PCR.

## Chlorophenol red β-D-galactopyranoside (CPRG) bacterial competition assay

To visualize and quantify the ability of *X. citri* to lyse *E. coli* strain MG1655, a CPRG-based method was used as described previously [40,55]. Briefly, to each well of a clear U-shaped bottom 96-well plate, 100 μL of a mixture of 0.5X buffer A (7.5 mM $(NH_4)_2SO_4$, 8.5 mM $Na_2HPO_4$, 11 mM $KH_2PO_4$, 25 mM NaCl, pH 7.0), 1.5% agarose and 40 μg/mL CPRG (Sigma-Aldrich) was added, and plates were thoroughly dried under a laminar flow. *X. citri* cells grown in the appropriate media, were mixed in a 1:1 volume ratio with a concentrated *E. coli* culture. The *E. coli* cultures were grown to $OD_{600} = 1$ in the presence of 0.2 mM IPTG (inducing the *lac* operon) in 2xYT medium, washed once and concentrated 10 times. Five microliters of *X. citri* and *E. coli* mixtures were immediately added to the 96-well plate without puncturing or damaging the agarose, covered with a transparent seal and quickly thereafter absorbance at 572 nm ($A_{572}$) was monitored over time in a 96-well plate reader for at least 200 minutes (SpectraMax Paradigm, Molecular Devices). The $A_{572}$ values were processed using RStudio software [56] and plotted using the ggplot2 package [57]. Background intensities obtained from the mean of $A_{572}$ values of unchallenged, non-lysing *E. coli* cells were subtracted from the data series and data were normalized for initial $OD_{600}$ differences.

## Fluorescence microscopy image acquisition and analysis

Briefly, 1 μL of cell suspension was spotted on a thin agarose slab containing 1X buffer A (15 mM $(NH_4)_2SO_4$, 17 mM $Na_2HPO_4$, 22 mM $KH_2PO_4$, 50 mM NaCl, pH 7.0) and 2% agarose and covered with a #1.5 cover glass (Corning). For time-lapse imaging, thicker agar slabs containing the appropriate media were constructed as described [26]. Phase contrast and msfGFP emission images were obtained with a Leica DMi-8 epifluorescent microscope. Fluorescence emiission of msfGFP were captured using 1000 to 1500 ms exposure times at maximum excitation light intensities. The microscope was equipped with a DFC365 FX camera (Leica), a HC PL APO 100x/1.4 Oil ph3 objective (Leica) and excitation-emission band-pass filter cubes for GFP (Ex.: 470/40, DC: 495, Em.: 525/50; Leica) foci, eleven 0.05 μm Z-plane stacks were obtained from a 0.5 μm region within the centre of the cells. This allowed for a better signal to noise ratio and increased detection of foci location in different depths of the cell. These image stacks were background subtracted by a rolling ball correction using a significant cell-free

portion of each image as a reference and, finally, combined by an average intensity projection using the FIJI software package [58]. To obtain a quantitative representation of cell sizes, background corrected fluorescence intensities and amount of foci present per cell, the images were analysed using the MicrobeJ software package [59] and data was analysed by RStudio software [56] and plotted using the ggplot2 package [57].

## Co-culture growth experiment

To illustrate the physiological burden associated with T4SS overexpression seven independent *X. citri Δ5´UTR$_{B7}$ virB11-msfgp* mutants and seven independent *X. citri virB11-msfgfp* strains (revertants to wild-type from the second recombination event), in two separate experiments, were grown overnight in 2xYT media, diluted 100-fold into defined AB media with 0.2% sucrose and 0.2% casamino acids and grown overnight. These overnight cultures were mixed 1:1 and 10-fold diluted into fresh AB media with 0.2% sucrose and 0.01% casamino acids. Immediately after mixing (at time point 0 h) fluorescence microscopy images were taken (as described above) to register the exact ratio of *X. citri virB11-msfgfp* cells versus *X. citri Δ5´UTR$_{B7}$ virB11-msfgfp* cells. The cultures were subsequently diluted regularly so to prevent cultures from reaching saturation which would halt further cell division. At the indicated timepoints several microscopy images of each of the co-cultures were again acquired. Given that the *virB11-msfgfp* reporter in the strain lacking the *5´UTR$_{B7}$* has a higher msfGFP production (histograms of the populations' msfGFP fluorescence levels do not overlap), cells could be sorted by using average msfGFP content and as such, an accurate quantification of the cell ratio between wild-type and deleted *5´UTR$_{B7}$* strains could be calculated over time.

## Transcription start site analysis

Transcriptional start sites of the *virD* and *virB* transcripts were identified using the 5´ RACE Kit (Roche) as described [27]. The oligonucleotide sequences for *virD4* and *virB7* 5´ RACE assays are listed in S1 Table. The resulting PCR fragments were blunt ligated into pGEM-T before sequencing 3 independent clones.

## RNA electrophoretic mobility shift assays

DNA fragments encoding either the entire 5´ UTR of the *virB* operon or a shortened fragment lacking the first 73 nucleotides, were amplified from *X. citri* genome using forward primers which include the T7 promoter sequence (see S1 Table). RNA transcripts of the cloned *5´UTR$_{B7}$* fragments were produced *in vitro* from the resulting purified PCR products by using the T7 Transcription Kit (Roche) and labelled by using the RNA 3´ End Biotinylation Kit (Pierce). Recombinant CsrA protein was purified as previously described [27]. Approximately 70 nM of purified CsrA protein and 6.25 nM Biotin-labeled RNA were mixed with binding buffer (10 mM HEPES (pH 7.3), 20 mM KCl, 1mM MgCl$_2$, 1 mM DTT, 5% glycerol, 0.1 μg/μL yeast tRNA, 20 U RNasin (Promega)) in a total reaction volume of 20 μL. The binding reactions were incubated at 25˚C for 20 min. A 5 μL aliquot of loading buffer (97% glycerol, 0.01% bromophenol blue, 0.01% xylene cyanol) was added to the binding reaction and immediately loaded and resolved by 5% native polyacrylamide gels. The binding assays and detection of RNA products were performed with the LightShift Chemiluminescent RNA EMSA Kit (Thermo Scientific). For the control reactions, 312.5 nM unlabelled (competitor) *virB 5´UTR$_{B7}$* RNA was added to the binding reactions.

## Western blot assays

Western blot assays were performed using total protein extract of wild-type and mutant *X. citri* strains as described previously [40]. Specific polyclonal antibodies against VirB7, VirB8, VirB9, VirB10 and X-Tfe$^{XAC2609}$ were used. In order to correct for sample loading variations, levels of the β subunit of RNA polymerase (RNApolβ) were also evaluated. Briefly, *X. citri* cells grown overnight in AB medium supplemented with 0.2% sucrose and 0.5% casamino acids were collected by centrifugation, resuspended, and quickly adjusted to the same optical density (OD$_{600nm}$ = 1.3). Cells from 1 ml culture were harvested by centrifugation for 5 min at 6000g, resuspended in 500 μL of 1XPBS plus 125 μL of denaturing sample buffer and incubated for 5 min at 95˚C. Samples were resolved in 15-well Tricine-SDS-PAGE gels [60], transferred to 0.2 μm nitrocellulose membranes (Bio Rad), and blocked for 60 min using 5% skimmed milk in 1XPBS. Primary antibodies produced in rabbit against VirB7 (AbVirB7; 1:20,000 dilution), VirB8 (AbVirB8; 1:10,000), VirB9 (AbVirB9; 1:4,000), VirB10 (AbVirB10; 1:5,000), X-Tfe$^{XAC2609}$ (AbXAC2609; 1:4,000) and in mouse against the β subunit of RNA polymerase from *E. coli* (AbRNApolβ; Abcam ab12087; 1:5,000) were used. Secondary goat anti-rabbit or anti-mouse IgG-AP conjugates (Bio Rad 1706518; 1:5,000) were used for AbVirB7, AbVirB8, AbXAC2609 and AbRNApolβ with BCIP (VWR 0885) and NBT (Sigma-Aldrich N6876) for protein detection, and secondary goat anti-rabbit IgG-IRDye 800CW (Li-Cor 32211; 1:8,000) was used for AbVirB9 and AbVirB10 before visualization using an Odyssey Imaging System (Li-Cor). Band intensities were quantified using the FIJI software package [58] and normalized with respect to the intensities obtained in the *X. citri* wild-type strain for each protein. Results represent the mean value from at least four experiments (minimum two biological replicates) and vertical lines the corresponding standard deviation.

## RNA and cDNA preparation

Bacterial total RNA was isolated from *X. citri* strains (wild-type and derivatives) using either TRIzol reagent (Thermo Scientific, 15596026) or RNAspin Mini Kit (GE Healthcare, 25-0500-71) according to the manufacturer´s instructions. Approximately 10$^8$ cells from an overnight culture in AB medium were used for each extraction. For cDNA first strand synthesis, 4 μg of total RNA were reverse transcribed using RevertAid First Strand cDNA Synthesis Kit (Thermo Scientific, K1622), according to the manufacturer's instructions and cDNA concentrations were subsequently adjusted to 0.1 μg/μl. The absence of contamination by genomic DNA in each sample was confirmed by means of PCR analysis using 16S-specific oligonucleotides.

## RT-qPCR and overlapping PCR assays

Real-time quantitative polymerase chain reaction (RT-qPCR) assays were performed using Maxima SYBR Green/ROX qPCR Master Mix (Thermo Scientific, K0222) according to manufacturer's instructions in a QuantStudio 3 Real-Time PCR System (Applied Biosystems). The oligonucleotide sequences used to analyse the expression of each gene can be found in S1 Table. Results were analysed with the QuantStudio Design and Analysis Software (Applied Biosystems) and processed using the ΔΔCt method as described [61] using the 16S gene as the internal reference and *X. citri* wild-type strain as the reference sample. Plotted values represent the mean from four RT-qPCR runs (two biological and two technical replicates for each gene), and vertical lines the accumulated error from standard deviations. Overlapping PCR assays were performed using Phusion High-Fidelity PCR Master Mix (Thermo Scientific, F531L) according to manufacturer's instructions. Nine pairs of oligonucleotides (named from I to IX) were employed (see S1 Table). The following templates were used: non-reverse transcribed *X. citri* RNA (negative amplification control), reverse transcribed *X. citri* RNA obtained as

described above (sample) and *X. citri* genomic DNA (positive amplification control). All PCR products were resolved in SYBR Safe (Invitrogen, S33102) stained 1% agarose gels. Overlapping PCR assays were performed in two biological replicates with similar results.

### GUS activity assay

To generate the translational *amy*::$P_{tac}$-5´$UTR_{B7}$-*gus* reporter fusion construct under control of the $P_{tac}$ constitutive promoter, virB7 5'-UTR plus its first two codons were first fused in-frame to the second codon of the promoterless *gusA* gene. The sequence containing the *virB7* 5'-UTR and its first two codons was amplified by PCR using the total DNA of the *X. citri* wild-type strain 306 as the template. The *gusA* coding sequence was amplified from the vector pBI121. The fragments described above were ligated by PCR and the resulting fragment containing the *5'UTR$_{B7}$-gus* fusion was cloned into the *BamHI* and *HindIII* of the pUC18-mini-Tn7T-LAC vector [62]. The resulting $P_{tac}$-5'$UTR_{B7}$-*gus* contruct was amplified by PCR and cloned between the *BamHI* and *XbaI* sites of the integrative vector pPM7g [54]. The primer sequences used in the above amplifications are listed in S1 Table. These constructs were used to transform the *X. citri* wild-type and/or *csrA* strains. Transformed colonies were selected on nutrient agar plus kanamycin. *X. citri* cells harboring the *amy*::$P_{tac}$-5´$UTR_{B7}$-*gus* reporter were cultured in nutrient broth medium for 20 h and assayed for β-glucuronidase (GUS) activity. Bacteria cells were diluted and disrupted in GUS assay buffer (20 mM Tris-HCl (pH 7.0), 10 mM 2-mercaptoethanol, 5 mM EDTA, 1% TritonX-100). GUS activities were determined after 1 h by measuring $A_{420nm}$ using PNPG (p-Nitrophenyl-β-D-glucuronide) as the substrate. GUS activity was expressed as $(1000 \times A_{420nm})/(\text{time in min} \times A_{600nm})$ in Miller units. Values presented are means ± standard deviations of three independent experiments.

### Supporting information

**S1 Fig. 5´RACE assay identifying the transcription start sites for *virD4* and *virB7*.** A) Bottom panel shows the two adjacent possible transcription start sites (TSS) of *virD4* (red and purple arrow) due to sequencing ambiguity. The possible start sites are indicated with a purple G and a red C in the nucleotide sequence (top panel). The *virD4* sequence starts at the end of the displayed nucleotide sequence. In between the TSS and *virD4* a putative open reading frame (ORF) coding for a protein of unknown function can be found. B) Similar analysis for the *virB7* with TSS depicted with red arrow (bottom panel) and red G in the nucleotide sequence (top panel). In parts A and B, translation start sites are depicted in red and open reading frames in blue. Putative RNA polymerase binding sites are shown in bold and putative ribosome binding sites are underlined in regular font.
(TIF)

**S2 Fig. *X. citri virB* operon transcriptional analysis.** The *upper panel* shows a diagram of *X. citri virB* operon, from *virB7* to *xac2609*. A black arrow indicates the promoter proposed for gene transcription, located upstream of *virB7*. Genes coloured in yellow correspond to structural T4SS components, in grey to hypothetical proteins, in green to X-Tfis, and in red to X-Tfes. Nine sets of primers (designated I to IX) amplifying overlapping regions were designed in order to determine the length of the polycistronic mRNA transcript (see Materials and Methods). The *lower panel* shows pictures of SYBR Safe stained 1% agarose gels in which the PCR products (depicted in the *upper panel*) were applied. C-: negative control, non-reverse transcribed *X. citri* RNA template; S: sample, reverse transcribed *X. citri* RNA template; C+: *X. citri* genomic DNA template.
(TIF)

**S3 Fig. Quantification of β-glucuronidase (GUS) activity using *amy*::*P_{tac}-5´UTR_{B7}-gus* reporter translational fusion constructs.** A) Nucleotide sequence of the 5'UTR_{B7} used in the *amy*::*P_{tac}-5´UTR_{B7}-gus* reporter (translational fusion of the promoterless *gusA* in-frame fused to the *5´UTR_{B7}* and first two codons of the *virB7* gene under control of the constitutive P_{tac} promoter). Single underline: nucleotides 1–72. Italic type: putative ribosome binding site (RBS) at positions 238–243. Red type: GGA motifs at positions +31, +42, +57, +70, +141 (bold), +194 and +238 (within RBS). Blue type: beginning of coding region for *gus* reporter gene in which the first two *virB7* codons were maintained. B) GUS activities of wild-type and *csrA* mutant cells harbouring the *amy*::*P_{tac}-5´UTR_{B7}-gus* reporter. C) GUS activities of *X. citri* wild-type cells harbouring the *amy*::*P_{tac}-5´UTR_{B7}-gus* reporter and its mutants. B7: wild-type *5´UTR_{B7}*. B7-d72: deletion of first 72 nucleotides of the *5´UTR_{B7}*. RBS-GAA: GGA to GAA mutation in the ribosome binding site (RBS) at position 238. 194-GAA: GGA to GAA at position 194. 141-GAA: GGA to GAA mutation at position 141. RBS/141-GAA: double mutant carrying both RBS-GAA and 141-GAA. 194/141-GAA, double mutant carrying both 141-GAA and 194-GAA. In B and C, Cells were cultured in nutrient both medium for 20 h and assayed for GUS activity. The quantification of GUS activity was performed using p-nitrophenyl β-D-glucuronide (PNPG) as substrate as described in Materials and Methods. Experiments were repeated three times. The means ± standard deviations are shown.
(TIF)

**S4 Fig. Electrophoresis mobility shift assay (EMSA).** EMSA shows direct *in vitro* binding of CsrA with the complete *5´UTR_{B7}* and a shortened Δnt1-73 *5´UTR_{B7}* fragment (lacking the first 73 nucleotides, up to and including the fourth GGA motif). See Fig 1 from the main text for the *5´UTR_{B7}* sequence. Binding reactions were performed using 70 nM of purified CsrA protein and 6.25 nM Biotin-labelled RNA. Addition of 312.5 nM unlabelled *5´UTR_{B7}* RNA competes with labelled RNA binding. See Materials and Methods for details.
(TIFF)

**S5 Fig. Whole nitrocellulose membranes employed for the preparation of Fig 2B.** Polyclonal specific antibodies (Ab) against VirB7, VirB8, VirB9, VirB10, and XAC2609 were used. In the case of the β subunit of RNA polymerase (RNApolβ), monoclonal antibodies were used (see Materials and Methods for details). Experiments were repeated five times for VirB7, VirB8, VirB9, VirB10, and XAC2609 and four times for RNApolβ showing similar results. Expected molecular weights for mature proteins are: VirB7 12.62 kDa, VirB8 37.38 kDa, VirB9 26.56 kDa, VirB10 41.53 kDa, XAC2609 47.10 kDa, and RNApolβ 154.20 kDa. Note that VirB10 is a proline-rich protein, therefore it presents an apparent higher molecular weight.
(TIFF)

**S6 Fig. Fluorescence distribution from the co-culture experiment of Fig 5.** Shown are the distributions of the mean fluorescence levels for the two co-culture experiments of strains X. *citri Δ5´UTRB7virB11-msfgfp* and *X. citri virB11-msfgfp* at the different sampling timepoints. A) Mean fluorescence distributions for the first experiment represented by red lines in Fig 5 at timepoints 0h, 6h30, 30h30, 72h and 168h. Fluorescence intensity distribution of the two strains measured separately at time-point 0h are also shown (*X. citri Δ5´UTRB7virB11-msfgfp* alone and *X. citri virB11-msfgfp* alone). The cut-off value used for cell-sorting at all timepoints, 800 rfu, is shown. B) Mean fluorescence distributions from the represented by green lines in Fig 5 at timepoints 0h, 6h, 30h, 54h,126h and 150h. Fluorescence intensity distribution of the two strains measured separately at timepoint 0h are also shown (*X. citri Δ5´UTRB7-virB11-msfgfp* alone and *X. citri virB11-msfgfp* alone). The cut-off value used for cell-sorting was 1100 rfu at all time points except for 30h where the cut-off used was 1500 rfu (indicated

with an *). This discrepancy at 30h was very likely due to altered microscopy settings on that day of measurement but did not impair the efficiency sorting cells in this experiment.
(TIFF)

**S1 Table. Primers, strains and plasmids used in this study.**
(PDF)

**S1 Movie. Time-lapse movie showing contact dependent lysis at the single-cell level during growth in media depleted for casamino acids (AB medium+ 0.2% sucrose + 0.01% casamino acids).** Movie starts after 25 hours of growth on the pad due to significantly reduced growth speeds. White arrows indicate regions were *E. coli* cells come into contact with the smaller sized *X. citri* cells. Scalebar: 5μm. Time stamp in bottom left of the movie. See Materials and Methods for details of growth conditions.
(AVI)

**S2 Movie. Time-lapse movie showing contact dependent lysis at the single-cell level during growth in rich media (AB medium + 0.2% sucrose + 0.2% casamino acids).** Movie was initiated in parallel with S1 Movie, starting at time point zero. White arrows indicate regions were *E. coli* cells come into contact with smaller sized *X. citri* cells. Scalebar: 5μm. Time stamp in bottom left of the movie. See Materials and Methods for details of growth conditions.
(AVI)

## Acknowledgments

We thank Alexandre Bruni-Cardoso for unlimited access to the Leica DMI8 microscope, Ioannis Passaris and Sander K. Govers for helpful discussions, suggestions and critical reading of the manuscript, and Gilberto Kaihami for the introduction to the R programming language.

## Author Contributions

**Conceptualization:** William Cenens, Chuck S. Farah.

**Data curation:** William Cenens, Chuck S. Farah.

**Formal analysis:** William Cenens, Chuck S. Farah.

**Funding acquisition:** Chuck S. Farah.

**Investigation:** William Cenens, Maxuel O. Andrade, Edgar Llontop, Cristina E. Alvarez-Martinez, Germán G. Sgro, Chuck S. Farah.

**Methodology:** William Cenens, Maxuel O. Andrade, Germán G. Sgro, Chuck S. Farah.

**Project administration:** Chuck S. Farah.

**Supervision:** Chuck S. Farah.

**Writing – original draft:** William Cenens, Chuck S. Farah.

**Writing – review & editing:** William Cenens, Maxuel O. Andrade, Edgar Llontop, Cristina E. Alvarez-Martinez, Germán G. Sgro, Chuck S. Farah.

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
