## [Decision Letter · Decision Letter 0]

1 Mar 2020

Dear Dr. Farah,

Thank you very much for submitting your manuscript "Bactericidal Type IV Secretion System Homeostasis in Xanthomonas citri" for consideration at PLOS Pathogens. As with all papers reviewed by the journal, your manuscript was reviewed by members of the editorial board and by several independent reviewers. The reviewers appreciated the attention to an important topic, but raised concerns and made suggestions for further improvement. Based on the reviews, we are likely to accept this manuscript for publication, providing that you modify the manuscript to fully address their remaining concerns. Please note that reviewer 3 has their comments uploaded as a separate file.

Sincerely,

Nian Wang

Guest Editor

PLOS Pathogens

Wenbo Ma

Section Editor

PLOS Pathogens

Kasturi Haldar

Editor-in-Chief

PLOS Pathogens

orcid.org/0000-0001-5065-158X

Michael Malim

Editor-in-Chief

PLOS Pathogens

orcid.org/0000-0002-7699-2064

Reviewer Comments (if any, and for reference):

Reviewer's Responses to Questions

**Part I - Summary**

Reviewer #1: In this manuscript, Farah and his colleagues report on a characterization of the type IV secretion system (T4SS) in Xanthomonas that functions to inject toxic effectors into neighboring bacteria. The authors characterize the role of the transcriptional regulator CsrA in controlling production of the T4SS by: i) identifying potential CsrA binding sites in the upstream untranslated region (5’UTR) of the virB operon, ii) showing the CsrA binds the UTR of the virB operon, iii) demonstrating that a csrA deletion and mutation of the 5’ untranslated region (UTR) confer elevated virB expression, and iv) evaluating the effects of various growth conditions on T4SS production. Finally, they present evidence that CsrA regulates production of T4SS nanomachines within a certain range regardless of the environmental conditions tested, and that overproduction of these machines confers a fitness cost. Overall, the manuscript is well conceived and the findings support the conclusions drawn. The significance is mitigated by the fact that CsrA has been shown to regulate the production of many other translocation systems, the study does not provide new mechanistic insights into our knowledge of how CsrA controls virB transcription and translation, and there appear to be unexplored factors and regulatory networks other then CsrA that main T4SS homeostasis under different environmental/growth conditions. Nevertheless, the work clearly shows that CsrA functions as a transcriptional regulator that acts on the 5’ UTR of the virB regulon to inhibit overproduction of the T4SS.

Reviewer #2: This manuscript reports a study of the regulatory function of CsrA (RsmA) on the type IV secretion machinery in Xanthomonas citri. All the experimental data shown in this manuscript clearly indicate that CsrA represses the expression of the T4SS and the accompanying cell-killing ability through the direct physical binding of the CsrA protein to the 5’UTR of virB7. Merits of this study include the employment of new techniques for better resolutions of the transcription data and improved visualization of the secretion machinery. Each experiment was designed in a very organized way and conducted rigorously. In spite of these merits, however, I could hardly find any new information that fills a knowledge gap in this area of science. The role of the T4SS in the bacterial competition of X. citri was previously discovered and the regulatory function of CsrA has been known in other bacterial systems. This study gives an impression that it was conducted to validate the previous microarray data through comprehensive experiments. If the authors want to publish this study in this journal, I strongly suggest that they clearly mention what point(s) of this study is the novel discovery that gives a scientific impact on the study of bacterial pathogens.

Reviewer #3: The manuscript is well written and presents very interesting data relating to the control of the Vir genes by CsrA. The experiments appear well thought out as to the role of CsrA in regulation of the vir genes. In Figure 2B the Western blots indicate the protein concentrations for the various Vir proteins. I wonder how specific the antibodies are in order to make a quantitative assessment. Although you quantify surface area based on cell counts, are you stating in figure captions that the cell counts were per strain? The results section seems to contain discussion points. Perhaps you can combine the results and discussion given discussion points in results. I am not sure the surface/T4SS data are critical to this paper although I have no objection to it remaining. As for the co-cultivation, it would be more interesting to look at in planta comparisons rather than in vitro. Growing for ~50 hr to see a difference seems to be a stretch. Minor comments on the pdf file.

**Part II – Major Issues: Key Experiments Required for Acceptance**

Reviewer #1: 1. Studies reported in figure 5 should be carried out using strains tagged with different fluorescent markers, e.g., GFP, mRFP.

2. Further studies aimed at localizing the CsrA binding site within the UTR, testing for CsrA binding to DNA, and monitoring CsrA cellular levels in different growth conditions would help to define CsrA's mechanism of action.

3. The physiology experiments should be repeated for the deltaUTR and deltaCsrA mutant strains.

Reviewer #2: N/A

Reviewer #3: (No Response)

**Part III – Minor Issues: Editorial and Data Presentation Modifications**

Reviewer #1: Comments:

1. Pg. 6. Can the authors exclude the possibility that CsrA binds the DNA corresponding to the 5’ UTR and not the RNA as a mechanism for controlling transcription or translation?

2. Pg. 6. Given that both the UTR’s of virD4 and virB encode proteins, it would seem important to knock these out with point mutations or small deletions to confirm that they don’t play a role in regulation.

3. Pg. 6. The deltaUTR mutation doesn’t phenocopy the deltaCsrA mutation with respect to the magnitude of the stimulatory effect, perhaps suggesting that CsrA might bind other regions in the promoter? What would the effect be of a double deletion? Also, a limited study evaluating effects of smaller UTR deletion mutations on gene expression could shed more light into the mechanism by which CsrA blocks gene expression. (At this point, there is very little experimentation exploring CsrA mechanism). Minimally, it would be straightforward to map the CsrA binding sit on the UTR RNA (and DNA if it binds DNA) using EMSAs.

4. Is it possible to quantitate CsrA production levels under different environmental/growth conditions – this could provide valuable insights into whether CsrA is acting directly or indirectly to regulate T4SS production.

5. Is virD4 subject to CsrA control? This seems to be an obvious question of importance to answer – in the absence of VirD4, the T4SS is presumably nonfunctional, implying there should be coordinate control over gene expression. Simple to address.

6. Pg. 9. L. 257. The nutrient limitation studies: what’s the broad conclusion, that there aren’t major changes in gene expression with changes in the nutrients/pH’s tested? This is a pretty limited outcome – corresponding tests with the DeltaCsrA and DeltaUTR variants would provide more insights into the role, if any, of CsrA on maintaining T4SS production levels under these different growth conditions.

7. Fig. 5. I really think different fluorescent tags are needed for these studies. Regardless of possible differences in the WT vs DeltaUTR in T4SS focal production, it’s difficult to believe that the two strains can be distinguished and sorted. This would be trivial with GFP and mRFP tags for example. It is also unknown whether mixing of strains under these different growth conditions may affect the overall morphologies and T4SS distributions in one or the other strain, further complicating efforts to distinguish the cells by sorting using a single fluorescent marker.

8. P. 13. L. 397. It is never shown that CsrA exerts the same repressing power over a wide range of growth conditions, especially since CsrA levels were never shown over the tested conditions. There may well be other regulators that kick in to govern T4SS production under different growth conditions, yielding a more complex picture of regulation than is currently envisioned.

9. P. 13. L. 402. It would be fairly straightforward to add some more insights about CsrA mechanism of action by testing for DNA binding, or binding to subregions of the UTR, monitoring CsrA levels in vivo to provide a direct test of the assumption that levels don’t change under different conditions, etc.

10. The figure legends should only convey information that enables the reader to interpret the figure, not describe the findings or provide conclusions - this is done in the Results and Discussion sections. Several of the figure legends should be revised to delete this redundant material.

Reviewer #2: N/A

Reviewer #3: Results seems to contain discussion to a limited extent.

PLOS authors have the option to publish the peer review history of their article (what does this mean?). If published, this will include your full peer review and any attached files.

Reviewer #1: No

Reviewer #2: No

Reviewer #3: No
---

## [Editor Report · Decision Letter 1]

18 Apr 2020

Dear Dr. Farah,

We are pleased to inform you that your manuscript 'Bactericidal Type IV Secretion System Homeostasis in Xanthomonas citri' has been provisionally accepted for publication in PLOS Pathogens.

Best regards,

Nian Wang

Guest Editor

PLOS Pathogens

Wenbo Ma

Section Editor

PLOS Pathogens

Kasturi Haldar

Editor-in-Chief

PLOS Pathogens

orcid.org/0000-0001-5065-158X

Michael Malim

Editor-in-Chief

PLOS Pathogens

orcid.org/0000-0002-7699-2064
---

## [Editor Report · Acceptance letter]

14 May 2020

Dear Dr. Farah,

We are delighted to inform you that your manuscript, "Bactericidal Type IV Secretion System Homeostasis in *Xanthomonas citri*," has been formally accepted for publication in PLOS Pathogens.

Best regards,

Kasturi Haldar

Editor-in-Chief

PLOS Pathogens

orcid.org/0000-0001-5065-158X

Michael Malim

Editor-in-Chief

PLOS Pathogens

orcid.org/0000-0002-7699-2064